# SPIRAL: SELF-SUPERVISED PERTURBATION-INVARIANT REPRESENTATION LEARNING FOR SPEECH PRE-TRAINING

**Wenyong Huang, Zhenhe Zhang, Yu Ting Yeung, Xin Jiang, Qun Liu**
Huawei Noah's Ark Lab
`{wenyong.huang,zhangzhenhe1,yeung.yu.ting}@huawei.com`
`{jiang.xin,qun.liu}@huawei.com`

## ABSTRACT

We introduce a new approach for speech pre-training named SPIRAL which works by learning denoising representation of perturbed data in a teacher-student framework. Specifically, given a speech utterance, we first feed the utterance to a *teacher* network to obtain corresponding representation. Then the same utterance is perturbed and fed to a *student* network. The student network is trained to output representation resembling that of the teacher. At the same time, the teacher network is updated as moving average of student's weights over training steps. In order to prevent representation collapse, we apply an in-utterance contrastive loss as pre-training objective and impose position randomization on the input to the teacher. SPIRAL achieves competitive or better results compared to state-of-the-art speech pre-training method wav2vec 2.0, with significant reduction of training cost (80% for BASE model, 65% for LARGE model). Furthermore, we address the problem of noise-robustness that is critical to real-world speech applications. We propose multi-condition pre-training by perturbing the student's input with various types of additive noise. We demonstrate that multi-condition pre-trained SPIRAL models are more robust to noisy speech (9.0% - 13.3% relative word error rate reduction on real noisy test data), compared to applying multi-condition training solely in the fine-tuning stage. Source code is available [1].

## 1 INTRODUCTION

Industrial-scale automatic speech recognition (ASR) systems are usually trained with ten-thousands of hours of hand-transcribed speech data (Galvez et al., 2021). However, labeling speech data is expensive and time-consuming, especially for languages with small speaker populations, or for specific domains (e.g., legal, financial, scientific).

Recently, methods of utilizing unlabeled speech data to improve speech recognition system have achieved remarkable progress. Amongst them, *self-training* (Manohar et al., 2015; Kahn et al., 2020a; Synnaeve et al., 2020a; Chen et al., 2020b; Xu et al., 2020; Park et al., 2020b; Xiao et al., 2021), also known as *pseudo-labeling*, starts by training an ASR model with labeled speech data, which is referred to as teacher model. Then the teacher model, usually combined with a language model (LM), is used to produce pseudo-labels for unlabeled speech data. Finally, the labeled data and the pseudo-labeled data are combined to train a new model, which is referred to as student model. The process is repeated by taking the student model as the teacher in next iteration. Another line of work is speech *pre-training* (van den Oord et al., 2019; Chung & Glass, 2020; Wang et al., 2020; Baevski et al., 2020b; Liu et al., 2020). Pre-training learns speech representation from unlabeled data in a self-supervised way. The pre-trained model is then fine-tuned on the labeled data. Self-training and pre-training are complementary as shown in recent work (Xu et al., 2021; Zhang et al., 2020).

In this paper, we introduce a new speech pre-training method which works by learning denoising representation of perturbed data with the teacher-student framework, named as Self-supervised

---

[1] https://github.com/huawei-noah/Speech-Backbones/tree/main/SPIRAL

Perturbation-Invariant Representation Learning (SPIRAL). Compared to state-of-the-art speech pre-training methods such as wav2vec 2.0 (Baevski et al., 2020b) and HuBERT (Hsu et al., 2021), our method allows end-to-end training with a single contrastive loss, and without relying on discrete unit discovery techniques such as vector quantization (Jégou et al., 2011; Baevski et al., 2020a;b) or iterative clustering process (Hsu et al., 2021). We apply multi-condition training with SPIRAL (Seltzer et al., 2013; Ko et al., 2015) to improve noise-robustness for the downstream speech tasks.

SPIRAL is motivated by the observation that human tolerates speech perturbations or distortions fairly well. For example, people can communicate effectively in a noisy environment, or over a distorted telephone channel. Therefore, we hypothesize that by learning representation invariant to perturbation, the model will learn high-level representation which can enhance speech applications.

To learn perturbation-invariant representation in a self-supervised way, we employ a teacher-student framework similar to Tarvainen & Valpola (2017). During pre-training, given a speech utterance, we guide the student network which consumes the perturbed utterance to learn from the teacher network which consumes the clean utterance. The student is trained to produce denoised representation of the perturbed utterance similar to teacher's representation of the clean utterance. Meanwhile, the teacher, which shares the same model architecture with student, is updated as moving average of the student's weights over past training steps.

We apply the *in-utterance* contrastive loss to avoid model collapse to trivial constant representation (Chopra et al., 2005). As speech utterance are sequential data, there is another possible trivial solution which we call *positional collapse*. Positional collapse occurs when the student "cheats" by exploiting position correlation in teacher's representation to minimize the loss, while ignoring the content of the input utterance. To prevent positional collapse, we propose *position randomization* by adding random number of paddings on both sides of input utterance to the teacher.

Large-scale speech pre-training is computationally demanding. To reduce computation cost, we adopt a gradual down-sampling strategy in SPIRAL model, which has been verified effective in speech recognition literatures with negligible performance degradation (Peddinti et al., 2018; Han et al., 2020b; Huang et al., 2020). We also speculate that aggressive down-sampling helps to remove redundancy in speech.

To evaluate the effectiveness of SPIRAL, we conduct experiments on LibriSpeech and Libri-Light datasets. By training a small convolutional classifier on the representation of a frozen SPIRAL model, we can achieve WER of 3.5% and 6.4% on Librispeech test-clean and test-other respectively. SPIRAL achieves competitive or better results compared to state-of-the-art speech pre-training methods, while being much more training-efficient. We also demonstrate that multi-condition pre-trained SPIRAL are more robust to noisy speech with 9.0% - 13.3% relative word error rate (WER) reduction on real noisy test data from ChiME-3 (Barker et al., 2015), compared to the model applying multi-condition training solely in fine-tuning stage.

## 2 RELATED WORK

*Mean Teacher* (MT) (Tarvainen & Valpola, 2017) proposes using a student network to learn from a teacher network which is the moving average version of the student in the semi-supervised learning setting. The authors apply a supervised loss for labeled data and a consistency loss between teacher and student predictions for unlabeled data. However, direct application of MT to self-supervised learning leads to representation collapse (Grill et al., 2020).

*Noisy student training* (NST) (Xie et al., 2020; Park et al., 2020b) is a self-training method. NST demonstrates the importance of the aggressive injection of noise into the student. Although not emphasized, no noise is injected into pseudo-labeling process of the teacher. We consider our work as an extension of self-training approach to the self-supervised learning regime. Instead of using the teacher to provide pseudo-labels, we utilize the teacher for pseudo-reference representation.

*Denoising autoencoders* (Vincent et al., 2008) learn to recover a clean input from a corrupted version. However, speech data contain redundancy which is irrelevant to some speech applications such as speech recognition. Previous work (Baevski et al., 2019) shows that speech pre-training by recovering masked input speech features is not effective. In SPIRAL, we instead enforce latent representation of a corrupted input to resemble that of the corresponding clean input.

*Bootstrap Your Own Latent* (BYOL) (Grill et al., 2020) is a self-supervised image representation learning method. The method is based on a teacher-student framework similar to MT. The authors refer to student network as online network and teacher network as target network. They observe that naive application of MT to self-supervised learning leads to trivial constant representation. They prevent the representation collapse by appending a predictor to the student network. The theory behind is under investigation (Chen & He, 2021; Tian et al., 2021). Our method draws inspirations from BYOL and shares the similar architecture, but there are crucial differences. Instead of learning a single global representation for an image as in BYOL, SPIRAL learns a sequence of representation for an utterance. We aim for sequence applications such as speech recognition. In our preliminary experiments, we observe that appending a predictor to student network is not sufficient to prevent trivial constant representation for sequential representation learning. We use in-utterance contrastive loss (Baevski et al., 2020b) combined with input position randomization to successfully avoid representation collapse. We still keep the predictor in SPIRAL, but only for the sake of performance improvement from our observation. Another difference is that BYOL does not perform representation denoising. BYOL applies perturbation, which they call augmentation, to both the inputs of the teacher and the student. We demonstrate that representation denoising is crucial for speech pre-training. When perturbation is applied to the teacher's input, the effectiveness of speech pre-training degrades drastically.

*Wav2vec 2.0* (Baevski et al., 2020b) is a self-supervised speech representation learning method which belongs to the masked prediction family. Masked prediction methods are effective for text pre-training (Devlin et al., 2019), but not for speech pre-training when naively applied (Baevski et al., 2019). The reason is that speech data contains redundancy such as speaker information, pronunciation variations, which are irrelevant to the semantic meaning of the utterance. To overcome this problem, wav2vec 2.0 perform masking in intermediate latent space and performs target discretization with a differentiable quantization scheme. However, quantization leads to a more complex model by introducing additional hyper-parameters and an additional diversity loss. SPIRAL does not utilize quantization, and still achieves competitive performance compared to wav2vec 2.0. We hypothesize that aggressive down-sampling and learning by matching output representation may help to remove redundancy from the learned representation. We leave the investigation of whether target discretization could further improve SPIRAL for future work.

Liang et al. (2018) demonstrates that under the supervised learning setting, enforcing noise-invariant representation by penalizing difference between clean and noisy data improves ASR model accuracy.

## 3 METHOD

### 3.1 SELF-SUPERVISED PERTURBATION-INVARIANT REPRESENTATION LEARNING (SPIRAL)

Figure 1 shows the diagram of SPIRAL in the pre-training stage, where we use two neural networks, a *student* $F_\theta$ and a *teacher* $F_{\theta'}$. The weights of the teacher $\theta'$ is the moving average of the weights of the student $\theta$. At step $t$, the weights of the teacher $\theta'_t$ are updated as

$$\theta'_t \leftarrow \alpha_t \theta'_{t-1} + (1 - \alpha_t)\theta_t, \tag{1}$$

where $\alpha_t$ determines the rate of weight updates. Given a speech utterance $\boldsymbol{X} = (\boldsymbol{x}_1, \ldots, \boldsymbol{x}_T)$ of length $T$, the student takes a perturbed version $\tilde{\boldsymbol{X}} = s(\boldsymbol{X}) = (\tilde{\boldsymbol{x}}_1, \ldots, \tilde{\boldsymbol{x}}_T)$ as input where $s(\cdot)$ is a perturbation function. The output of the student is a representation sequence $\boldsymbol{Z} = F(\tilde{\boldsymbol{X}}; \theta) = (\boldsymbol{z}_1, \ldots, \boldsymbol{z}_T)$. The teacher takes the same utterance without perturbation as input and output another representation sequence $\boldsymbol{Z}' = F(\boldsymbol{X}; \theta') = (\boldsymbol{z}'_1, \ldots, \boldsymbol{z}'_T)$. For each representation $\boldsymbol{z}_i \in \boldsymbol{Z}$, the student is trained to match the teacher's representation $\boldsymbol{z}'_i$ at the same position amongst $k$ distracting samples. The distracting samples are randomly drawn from other positions of the same utterance in $\boldsymbol{Z}'$, which is found to be more effective than samples drawn from an entire batch of utterances (Baevski et al., 2020b). The in-utterance contrastive loss is defined following Sohn (2016); Wu et al. (2018) as,

$$\mathcal{L} = -\sum_{i=1}^{T} \log \frac{\exp(\phi(\boldsymbol{z}_i, \boldsymbol{z}'_i)/\kappa)}{\sum_{j \in D_i} \exp(\phi(\boldsymbol{z}_i, \boldsymbol{z}'_j)/\kappa)}, \tag{2}$$

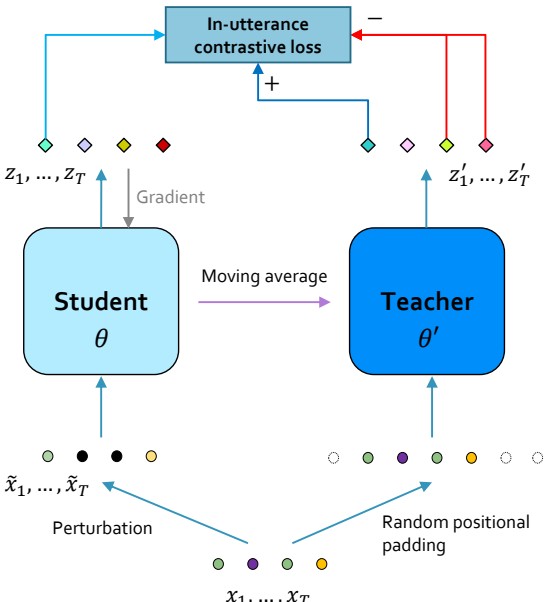

Figure 1: Illustration of SPIRAL architecture for speech pre-training.

where $\phi(\mathbf{a}, \mathbf{b}) = \mathbf{a}^T \mathbf{b} / \|\mathbf{a}\| \|\mathbf{b}\|$ is cosine similarity, $D_i$ is the set of indices of distractors for the $i$-th position, and $\kappa$ is the temperature parameter.

However, applying in-utterance contrastive loss could cause a kind of representation collapse which we refer to as positional collapse. Contrastive candidates are sampled based on their positions in utterances. When a teacher's representation $z_i'$ is correlated with its position $i$ (e.g., correlation introduced by positional encoding in Transformer), the student could exploit this correlation to generate its representation $z_i$ solely based on the position index $i$, while ignoring content of the input. In this case, the model does not learn meaningful representation of the input content. Therefore, we prevent positional collapse by randomizing positions of teacher's representation. In particular, we add random number of padding data at both ends of the input to the teacher to randomly shift the position information for each output representation $z_i'$. The student thereby is unable to exploit the spurious position information to minimize the contrastive loss. Note that when calculating the contrastive loss, we exclude the corresponding representation of the padded data.

## 3.2 MODEL ARCHITECTURE

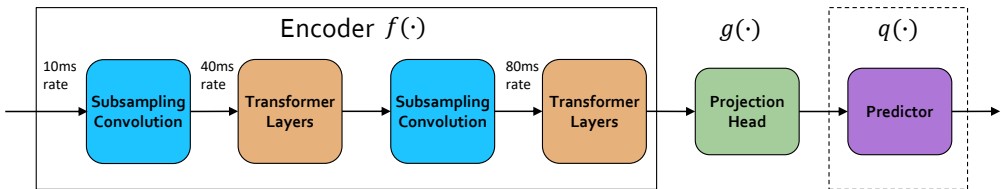

Figure 2: The architecture of the student model in SPIRAL. The frame rate of input is denoted as '10/40/80 ms'. The dashed line indicates the optional predictor which can be removed with small performance degradation. The structure of the teacher model is the same but without the predictor.

As illustrated in Figure 2, student $F_\theta$ is composed of an encoder $f(\cdot)$, a projection head $g(\cdot)$ (Chen et al., 2020a) and an optional predictor $q(\cdot)$ (Grill et al., 2020), i.e., $F_\theta = (f \circ g \circ q)(\cdot; \theta)$. The teacher $F_{\theta'}$ has the same structure expect that it has no predictor, $F_{\theta'} = (f \circ g)(\cdot; \theta')$. The encoder consists of two blocks. In each block, we first apply temporal convolutions to perform down-sampling, followed by Transformer (Vaswani et al., 2017) with convolutional relative position encoding (Baevski et al.,

2020b). Each convolution is followed by layer normalization (LN) (Ba et al., 2016) and ReLU. For the projection head, we apply a simple linear layer. The predictor consists of two layers of temporal convolution and a linear layer. The convolutions are followed by batch normalization (BN) (Ioffe & Szegedy, 2015) and ReLU.

During pre-training, we add computation noise to both the student and the teacher by applying dropout (Srivastava et al., 2014) and LayerDrop (Fan et al., 2020) in Transformer. We use the same dropout and LayerDrop rates for the student and the teacher.

### 3.3 ADAPTIVE SPECAUGMENT

We apply adaptive SpecAugment similar to Park et al. (2020a) as the primary perturbation method. Along either time or frequency dimension, we sample uniformly a certain proportion $p$ of all time-steps to be start indices and mask the subsequent consecutive $L$ time-steps. The masked time-steps is filled with zeros along frequency dimension. Along time dimension, we use Gaussian noise as masking values to avoid numerical problems for LN (Park et al., 2020a).

### 3.4 MULTI-CONDITION PRE-TRAINING

For noise-robust pre-training with SPIRAL, we perturb input of the student with various types of additive noise. We consider this technique as an implementation of multi-condition training (MCT) (Seltzer et al., 2013) in self-supervised setting. Specifically, for each input utterance to the student, we sample noise clips from a noise dataset, and mix the noise clips with the whole utterance by addition in time-domain. We first uniformly sample a signal-to-noise ratio (SNR) from a pre-defined range for each utterance. Then we scale the noise volume according to the required SNR. In our preliminary experiments, we found that applying additive noise alone as perturbation degrades performance. Therefore, we apply additive noise perturbation together with adaptive SpecAugment.

### 3.5 MODEL FINE-TUNING

After pre-training, we take the encoder from the student model in SPIRAL and add a randomly initialized convolutional classifier on top of it. The convolutional classifier is composed of two layers of convolution, followed by LN and ReLU, and a linear output projection. The convolution filters consist of 512 channels with kernel width of 5.

We fine-tune the model with connectionist temporal classification (CTC) (Graves et al., 2006) objective for speech recognition. We use 1024 subwords as output units. The sub-words are generated from training transcripts of LibriSpeech with SentencePiece (Kudo & Richardson, 2018).

We further investigate SPIRAL's ability to learn high-level representation of speech during pre-training. In addition to whole-model fine-tuning, we apply frozen fine-tuning. We freeze the pre-trained parameters and only fine-tune the convolutional classifier which can only perform local classification due to limited receptive field.

## 4 EXPERIMENTAL SETUP

### 4.1 DATA

For pre-training, we use the 960-hour training data (ignoring the labels) from LibriSpeech (Panayotov et al., 2015)(LS-960), or 60k-hour unlabeled audio data from Libri-Light (Kahn et al., 2020b) (LL-60K). For Libri-Light, we segment the data using official tools with a threshold of 16s, resulting in 46.9k hours of data. The two datasets are both derived from English audiobooks from LibriVox project[2]. For ASR fine-tuning, we apply 100-hour subset (train-clean-100) as low-resource labeled data and entire LS-960 with labels as high-resource labeled data, both from LibriSpeech.

For multi-condition training, we use the noise dataset from Reddy et al. (2021). The dataset consists of 181 hours of noise data with about 150 noise types and 70,000 clips. We shuffle and split the noise data with a ratio of 8:1:1, which are used for training, synthesizing noisy dev-sets and synthetic noisy

---

[2]https://librivox.org/

Table 1: Detailed configurations of the SPIRAL BASE and LARGE models.

| Modules | Conv.1 | Transf.1 | Conv.2 | Transf.2 | Proj. H. | Predictor | #Params |
|---|---|---|---|---|---|---|---|
| Hyper-params | kernel size channel stride | layer emb. dim. ffn dim. layerdrop attn. heads | kernel size channel stride | layer emb. dim. ffn dim. layerdrop attn. heads | dim. | kernel size channel | |
| BASE model | 5,5,1 384,512,512 2,2,1 | 2 512 2048 0 8 | 5,1 1536,768 2,1 | 10 768 3072 0.05 12 | 256 | 5,5,1 256,256,256 | 91.5M |
| LARGE model | 5,5,1 384,512,512 2,2,1 | 4 512 2048 0.05 8 | 5,1 2048,1024 2,1 | 20 1024 4096 0.05 16 | 512 | 5,5,1 512,512,512 | 287M |

Table 2: Comparison of pre-training cost between wav2vec 2.0 and SPIRAL.

| Model | Unlabeled data | Training steps | GPU days | Mixed precision |
|---|---|---|---|---|
| Wav2vec 2.0 BASE (Baevski et al., 2020b) | LS-960 | 500k | 102.4 | ✓ |
| SPIRAL BASE | LS-960 | 200k | 20.8 | - |
| Wav2vec 2.0 LARGE (Baevski et al., 2020b) | LL-60k | 1000k | 665.6 | ✓ |
| SPIRAL LARGE | LL-60k | 500k | 232.0 | - |

test-sets (results in Appendix A.2) respectively. SNRs of speech mixtures are set from 0 to 30 dB. We evaluate on real noisy data test set from CHiME-3 (Barker et al., 2015), which is comprised of speech data recorded in real noisy environments (bus, cafe, pedestrian area, and street junction). The data are recorded with a microphone array composed of multiple microphone channels located at different positions of a tablet, and a close-talking microphone.

## 4.2 TRAINING SETUPS

We apply 128-dimensional log-mel filterbank extracted with 20 ms window and 10 ms stride as the input acoustic feature. We experiment with BASE model and LARGE model configurations as shown in Table 1. The numbers of parameters are comparable to wav2vec 2.0 BASE and LARGE models correspondingly. For SpecAugment, we set $p = 0.025$ and $L = 20$ for time-dimension mask, and $p = 0.02$ and $L = 20$ for frequency-dimension mask.

In pre-training, we optimize with Adam (Kingma & Ba, 2015) optimizer, warming up the learning rate for the first 8% of updates to a peak of 3e-3. Then the learning rate decays to 0 with a cosine schedule. The moving average update rate $\alpha_t$ of teacher's weight also follows a cosine schedule (Grill et al., 2020). We increase $\alpha_t$ from 0.995 to 1.0 and from 0.990 to 0.999 for BASE and LARGE models respectively. We train the BASE model with batch size of 24 per GPU for 200k steps on 16 V100 GPUs, which takes about 1.3 days. For the LARGE model, we train with batch size of 20 per GPU for 500k steps on 32 V100 GPUs, which takes about 7.25 days. As shown in Table 2, there is a significant reduction of training cost (GPU days) compared to wav2vec 2.0 (Baevski et al., 2020b). SPIRAL requires 80% and 65% less training cost for BASE and LARGE respectively. Note that mix-precision training is not applied for SPIRAL yet.

For fine-tuning, we optimize with Adam and a tri-state rate schedule where the learning rate is warmed up for the first 10% of updates to 3e-5, held constant for the next 40% and then linearly decayed to zero following Baevski et al. (2020b). We fine-tune BASE and LARGE with batch size of 14 and 18 per GPU respectively on 8 GPUs for 80k steps on train-clean-100. We fine-tune LARGE with batch size of 10 per GPU on 16 GPUs for 320k steps on LS-960. We apply SpecAugment for whole-model fine-tuning but not for frozen fine-tuning. For multi-condition pre-training and

Table 3: ASR results fine-tuned from low-resource train-clean-100. Language models used in decoding are listed in LM. We compare SPIRAL BASE pre-trained on LS-960 and SPIRAL LARGE pre-trained on LL-60k with previous methods. We report WER (%) on Librispeech dev/test sets.

| Model | Unlabeled data | LM | dev | | test | |
|---|---|---|---|---|---|---|
| | | | clean | other | clean | other |
| **Supervised/Semi-Supervised** | | | | | | |
| Hybrid DNN/HMM (Lüscher et al., 2019) | - | 4-gram | 5.0 | 19.5 | 5.8 | 18.6 |
| Iter. pseudo-labeling (Xu et al., 2020) | LL-60k | 4-gram+Transf. | 3.19 | 6.14 | 3.72 | 7.11 |
| Noisy student (Park et al., 2020b) | LS-860 | LSTM | 3.9 | 8.8 | 4.2 | 8.6 |
| **Self-supervised** | | | | | | |
| wav2vec 2.0 BASE (Baevski et al., 2020b) | LS-960 | - | 6.1 | 13.5 | 6.1 | 13.3 |
| SPIRAL BASE frozen (ours) | LS-960 | - | 7.9 | 12.7 | 7.6 | 13.0 |
| SPIRAL BASE (ours) | LS-960 | - | 5.5 | 11.1 | 5.4 | 11.2 |
| wav2vec 2.0 BASE (Baevski et al., 2020b) | LS-960 | 4-gram | 2.7 | 7.9 | 3.4 | 8.0 |
| SPIRAL BASE (ours) | LS-960 | 4-gram | 2.7 | 7.0 | 3.3 | 7.5 |
| wav2vec 2.0 BASE (Baevski et al., 2020b) | LS-960 | Transf. | 2.2 | 6.3 | 2.6 | 6.3 |
| SPIRAL BASE (ours) | LS-960 | Transf. | 2.3 | 5.8 | 2.7 | 6.1 |
| wav2vec 2.0 LARGE (Baevski et al., 2020b) | LL-60k | - | 3.3 | 6.5 | 3.1 | 6.3 |
| SPIRAL LARGE frozen (ours) | LL-60k | - | 7.1 | 9.2 | 6.6 | 9.7 |
| SPIRAL LARGE (ours) | LL-60k | - | 3.3 | 5.9 | 3.3 | 6.3 |
| wav2vec 2.0 LARGE (Baevski et al., 2020b) | LL-60k | Transf. | 1.9 | 4.0 | 2.0 | 4.0 |
| SPIRAL LARGE (ours) | LL-60k | Transf. | 1.9 | 3.9 | 2.2 | 4.3 |

fine-tuning, we randomly perturb each utterance with additive noise with 50% probability before applying SpecAugment. SNR is uniformly sampled from 0-30 dB.

### 4.3 LANGUAGE MODEL AND DECODING

We use a word-level Transformer LM (Baevski & Auli, 2019) trained on Librispeech LM corpus which is identical to Synnaeve et al. (2020b). For low-resource ASR setting, we also evaluate SPIRAL BASE with the official LibriSpeech 4-gram LM. We observe that models fine-tuned with subword units performs worse than models fine-tuned with character units when decoding with word-level LM. Therefore, we apply character-based models for LM decoding, which is the same setting as wav2vec 2.0. The results of LM decoding with subword-based models are available in Appendix A.1.

As output frame rate of pre-trained SPIRAL encoder is low (80ms), the output sequence may be too short for character units. To reuse the pre-trained encoder, we devise an upsampling strategy for the SPIRAL encoder output in fine-tuning stage. We apply a 1-D convolution layer to project the original encoder output of dimension $d$ into a vector of dimension $4d$. At each time-step, we reshape the projected output vector from $(1, 4d)$ to $(4, d)$. The frame rate now becomes 20ms. Then we feed the upsampled outputs to convolutional classifier.

We perform random search for decoding parameters and choose the best parameters according to performance on dev-other with beam 50. The final test performance is measured with beam 500. We use the beam search decoder of Pratap et al. (2019).

## 5 RESULTS

### 5.1 EVALUATION UNDER LOW-RESOURCE AND HIGH-RESOURCE LABELED DATA SETTINGS

We first evaluate our method under a low-resource ASR setting in which we fine-tune the models with 100-hour LibriSpeech data (train-clean-100). The results are shown in Table 3. We evaluate a BASE model pre-trained with 960-hour LibriSpeech (LS-960) and a LARGE model pre-trained with Libri-Light (LL-60K). The frozen BASE model performs well, achieving a WER of 13.0%

Table 4: ASR results fine-tuned from high-resource LS-960. Language models used in decoding are listed in LM. We compare SPIRAL LARGE pre-trained on Libri-Light (LL-60k) with previous methods. We report WER (%) on Librispeech dev/test sets.

| Model | Unlabeled data | LM | dev | | test | |
|---|---|---|---|---|---|---|
| | | | clean | other | clean | other |
| **Supervised** | | | | | | |
| ContextNet (Han et al., 2020a) | - | LSTM | 1.9 | 3.9 | 1.9 | 4.1 |
| Conformer (Gulati et al., 2020) | - | LSTM | 2.1 | 4.3 | 1.9 | 3.9 |
| **Semi-supervised** | | | | | | |
| CTC Transf. + PL (Synnaeve et al., 2020a) | LL-60k | CLM+Transf. | 2.10 | 4.79 | 2.33 | 4.54 |
| S2S Transf. + PL (Synnaeve et al., 2020a) | LL-60k | CLM+Transf. | 2.00 | 3.65 | 2.09 | 4.11 |
| Iter. pseudo-labeling Xu et al. (2020) | LL-60k | 4-gram+Transf. | 1.85 | 3.26 | 2.10 | 4.01 |
| Noisy student (Park et al., 2020b) | LL-60k | LSTM | 1.6 | 3.4 | 1.7 | 3.4 |
| **Self-supervised** | | | | | | |
| wav2vec 2.0 LARGE (Baevski et al., 2020b) | LL-60k | - | 2.1 | 4.5 | 2.2 | 4.5 |
| SPIRAL LARGE frozen (ours) | LL-60k | - | 4.0 | 6.2 | 3.5 | 6.4 |
| SPIRAL LARGE (ours) | LL-60k | - | 2.1 | 4.3 | 2.2 | 4.6 |
| wav2vec 2.0 LARGE (Baevski et al., 2020b) | LL-60k | Transf. | 1.6 | 3.0 | 1.8 | 3.3 |
| SPIRAL LARGE (ours) | LL-60k | Transf. | 1.5 | 3.1 | 1.8 | 3.5 |

on test-other, which is on par with wav2vec 2.0 BASE. This suggests that SPIRAL indeed learns meaningful high-level representations in a self-supervised way. When we fine-tune the whole BASE model, the model achieves WER of 5.4% and 11.2% on test-clean and test-other respectively, outperforming wav2vec 2.0 BASE with 11.5% and 15.8% relative WER reduction. When decoding with Transformer LM, the BASE model achieves WER of 2.7% and 6.1% on test-clean and test-other respectively. The results are on par with wav2vec 2.0 BASE.

The SPIRAL LARGE model consists of more parameters and is pre-trained with more data. The model achieves WER of 2.2% and 4.3% on test-clean and test-other respectively. The significant improvement of LARGE over BASE demonstrates the scalability of SPIRAL. The results of SPIRAL LARGE are competitive to wav2vec 2.0 LARGE. This is encouraging, as SPIRAL LARGE only takes 35% of training cost of wav2vec 2.0 LARGE.

We further evaluate SPIRAL LARGE pre-trained with Libri-Light (LL-60K) under a high-resource ASR setting with 960-hour LS-960 as fine-tuning data. As shown in Table 4, the LARGE model achieves WER of 1.8% and 3.5% on test-clean and test-other respectively, which are on par with the wav2vec 2.0 LARGE model. We note that the supervised models and the noisy student model (Park et al., 2020b) in Table 4 are autoregressive models. Our models are fine-tuned with CTC objective which is non-autoregressive and generally inferior to autoregressive models. We use CTC objective for its simplicity and comparability to previous speech pre-training methods.

We consider SPIRAL as a preferred alternative to wav2vec 2.0 given that SPIRAL only requires 20%−35% computation cost of wav2vec 2.0. We expect further efficiency improvement when we implement mix-precision training for SPIRAL.

## 5.2 NOISE-ROBUST PRE-TRAINING

To evaluate noise-robustness of the pre-trained models, we compare the effects of applying multi-condition training (MCT) in pre-training or fine-tuning stages of SPIRAL. The results are shown in Table 5. The vanilla SPIRAL BASE model and wav2vec 2.0 BASE model deteriorate with significantly higher WER on noisy test data.

On real noisy test speech data in CHiME-3 for different microphone channels (ch), SPIRAL with multi-condition pre-training significantly improves speech recognition performance. Compared to the model applying MCT solely in fine-tuning, applying MCT both in pre-training and fine-tuning achieves 12.4%, 13.3% and 9.0% relative WER reduction for ch 1, 5 and 2 respectively. There is smaller performance improvement of 3.8% relative WER reduction for ch 0, which is a close-talking microphone with the highest SNR. We note that ch 2 faces backwards to the speaker. SNR

Table 5: Evaluation on noise-robustness of the models. We use wav2vec 2.0 BASE released by the authors as the baseline. The SPIRAL BASE models are pre-trained with LS-960 and fine-tuned with train-clean-100. We report WER (%) on Librispeech and CHiME-3 real data test sets.

| BASE model | Pre-train w/ MCT | Fine-tune w/ MCT | Librispeech | | CHiME-3 | | | |
|---|---|---|---|---|---|---|---|---|
| | | | clean | other | ch0 | ch5 | ch1 | ch2 |
| wav2vec 2.0 | - | - | 6.1 | 13.3 | 23.2 | 56.1 | 68.3 | 98.1 |
| SPIRAL | - | - | 5.4 | 11.2 | 24.1 | 52.1 | 58.9 | 92.6 |
| SPIRAL | - | ✓ | 5.7 | 11.4 | 20.8 | 35.5 | 41.1 | 76.4 |
| SPIRAL | ✓ | - | 5.7 | 11.5 | 20.8 | 33.6 | 38.5 | 74.0 |
| SPIRAL | ✓ | ✓ | 5.9 | 11.4 | 20.0 | 31.1 | 35.6 | 69.5 |

of the recordings from ch 2 is the lowest, leading to high WER. We note that other pre-training methods including wav2vec 2.0 may benefit from multi-condition training, which are worth for further investigation.

## 5.3 ABLATIONS

### 5.3.1 INPUT PERTURBATION AND COMPUTATION NOISE OF TEACHER

SPIRAL learns denoised representation of perturbed data. By default, we only apply perturbation to the input of the student. An alternative method is to perturb both inputs of the teacher and the student, and optimize consistency between their representations (Grill et al., 2020; Chen & He, 2021). We conduct experiments to evaluate the effects of perturbing the input and adding computation noise (dropout and LayerDrop) to the teacher. The results are shown in Table 8 in Appendix A.3. The results suggest that applying SpecAugment to teacher's input degrades performance significantly. Performance degradation decreases but is still significant with lower ratio and width of the masks. This supports the necessity of representation denoising, and our view of SPIRAL as an extension of self-training in which teacher network are fed with clean input. The results also support applying computation noise to teacher during pre-training. There is a 15.9% relative WER reduction with computation noise. This may be linked to Gal & Ghahramani (2016).

### 5.3.2 EFFECTS OF PREDICTOR AND PROJECTION HEAD

We do ablation studies to understand the role of predictor and projection head in SPIRAL. The results are shown in Table 9 in Appendix A.3. When removing the predictor from the student, we observe performance degradation, but representation collapse does not happen. In the architectures relying on predictor to prevent collapse (Grill et al., 2020; Chen & He, 2021), applying batch normalization (BN) in the predictor is essential. While in SPIRAL, we observe that BN in the predictor can be replaced by layer normalization (LN) with a small performance degradation. When the predictor is removed, we observe performance improvement by applying a convolutional projection head. The convolutional projection head is composed of a temporal convolution layer with LN and ReLU, and a linear layer. But applying convolutional projection head to the model with a predictor, there is no further performance improvement. This suggests that convolutional projection head and predictor play a similar role in SPIRAL, and they are not complementary.

## 6 CONCLUSION

We presented SPIRAL, a new approach to speech pre-training by learning denoising representation of perturbed data with a teacher-student framework. SPIRAL can learn high-level speech representation in self-supervised way. Training a small convolutional classifier on frozen representation of SPIRAL achieves WER of 3.5% and 6.4% on Librispeech test-clean and test-other respectively. We show that SPIRAL achieves competitive or better results compared to state-of-the-art speech pre-training methods, with significant reduction of training cost. We investigate multi-condition pre-training and demonstrates that multi-condition pre-training is more effective than solely applying multi-condition training in the fine-tuning stage. We presume SPIRAL as a general pre-training method, which can apply to other modalities such as images and text. We leave it for future work.

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

# A APPENDIX

## A.1 OUTPUT UNITS COMPARISON FOR LANGUAGE MODEL DECODING

We evaluate decoding performance of different combinations of SPIRAL output units and Transformer LM. The subword-level LM is trained on the same Librispeech LM corpus and shares the same 1024 subwords as the SPIRAL model fine-tuned with subword units.

Table 6: ASR results fine-tuned from low-resource train-clean-100 and high-resource train-960. The model units and language models for decoding are listed in fine-tuning units and LM respectively. We compare SPIRAL BASE pre-trained on LS-960 and SPIRAL LARGE pre-trained on LL-60k with previous methods. We report WER (%) on Librispeech dev/test sets.

| Model | Unlabeled data | Fine-tuning units | LM | dev | | test | |
|---|---|---|---|---|---|---|---|
| | | | | clean | other | clean | other |
| **Low-resource** | | | | | | | |
| SPIRAL BASE | LS-960 | subword | - | 5.5 | 11.1 | 5.4 | 11.2 |
| SPIRAL BASE | LS-960 | char | - | 5.3 | 11.0 | 5.4 | 11.1 |
| SPIRAL BASE | LS-960 | subword | word | 2.9 | 6.8 | 3.2 | 7.2 |
| SPIRAL BASE | LS-960 | subword | subword | 2.7 | 6.3 | 2.9 | 6.7 |
| SPIRAL BASE | LS-960 | char | word | 2.3 | 5.8 | 2.7 | 6.1 |
| SPIRAL LARGE | LL-60k | subword | - | 3.3 | 5.9 | 3.3 | 6.3 |
| SPIRAL LARGE | LL-60k | char | - | 3.5 | 6.3 | 3.5 | 6.7 |
| SPIRAL LARGE | LL-60k | subword | word | 2.4 | 4.5 | 2.5 | 4.8 |
| SPIRAL LARGE | LL-60k | subword | subword | 2.3 | 4.5 | 2.4 | 4.8 |
| SPIRAL LARGE | LL-60k | char | word | 1.9 | 3.9 | 2.2 | 4.3 |
| **High-resource** | | | | | | | |
| SPIRAL LARGE | LL-60k | subword | - | 2.1 | 4.3 | 2.2 | 4.6 |
| SPIRAL LARGE | LL-60k | char | - | 2.2 | 4.5 | 2.3 | 4.7 |
| SPIRAL LARGE | LL-60k | subword | word | 1.7 | 3.5 | 1.9 | 3.7 |
| SPIRAL LARGE | LL-60k | subword | subword | 1.6 | 3.3 | 1.7 | 3.5 |
| SPIRAL LARGE | LL-60k | char | word | 1.5 | 3.1 | 1.8 | 3.5 |

## A.2 PERFORMANCE ON SYNTHETIC NOISY DATASET

On the synthetic noisy dataset (NS-Librispeech) with matched SNR range (0-30 dB) of the training data, SPIRAL pre-trained and fine-tuned with MCT is more effective than applying MCT solely in fine-tuning. We observe 5.3% and 9.1% relative WER reduction on synthetic noisy test-clean and test-other sets respectively.

Table 7: Evaluation on noise-robustness of the models. We use wav2vec 2.0 BASE released by the authors as the baseline. The SPIRAL BASE models are pre-trained with LS-960 and fine-tuned with train-clean-100. We report WER (%) on Librispeech test sets and synthetic noisy Librispeech test sets at 0 - 30 dB (NS-Librispeech).

| BASE model | Pre-train w/ MCT | Fine-tune w/ MCT | Librispeech | | NS-Librispeech | |
|---|---|---|---|---|---|---|
| | | | clean | other | clean | other |
| wav2vec 2.0 | - | - | 6.1 | 13.3 | 14.4 | 27.4 |
| SPIRAL | - | - | 5.4 | 11.2 | 12.2 | 23.3 |
| SPIRAL | - | ✓ | 5.7 | 11.4 | 7.6 | 16.5 |
| SPIRAL | ✓ | - | 5.7 | 11.5 | 7.4 | 15.8 |
| SPIRAL | ✓ | ✓ | 5.9 | 11.4 | 7.2 | 15.0 |

## A.3 RESULTS OF ABLATION STUDIES

Here are the results of ablation studies discussed in Section 5.3.

Table 8: Ablation studies of input perturbation with SpecAugment and computation noise on teacher. We list the mask ratio and mask length as $p$, $L$ for time and frequency masks. The first row is the default setting of SPIRAL. We apply SPIRAL BASE fine-tuned with train-clean-100, and report WER (%) on the Librispeech dev-other set.

| Time mask | Frequency mask | Computation noise | dev other |
|-----------|----------------|-------------------|-----------|
| - | - | ✓ | 11.1 |
| - | - | - | 13.2 |
| 0.025, 20 | 0.02, 20 | ✓ | 47.9 |
| 0.0125, 20 | 0.01, 20 | ✓ | 42.8 |
| 0.025, 10 | 0.02, 10 | ✓ | 39.4 |

Table 9: Ablation studies of predictor and projection head in SPIRAL. We apply SPIRAL BASE fine-tuned with train-clean-100, and report WER (%) on the Librispeech dev-other set.

| Architecture | dev other |
|--------------|-----------|
| SPIRAL BASE | 11.1 |
| + predictor use LN | 11.6 |
| + conv proj. head | 11.5 |
| – predictor | 13.7 |
|    + conv proj. head | 12.1 |

