# OpenReview forum: "SPIRAL: Self-supervised Perturbation-Invariant Representation Learning for Speech Pre-Training"
_ICLR.cc/2022/Conference — ICLR 2022 Poster_

### Official Review · Reviewer_J87m · 2021-11-01

**Correctness:** 3
**Technical Novelty And Significance:** 2
**Empirical Novelty And Significance:** 4
**Recommendation:** 6
**Confidence:** 4

**Main Review:**

Strengths:
1. While the paper adapts the teacher-student self-supervised pre-training framework that has been studied for image representation learning (Grill et al., Chen & He, Tian et al.), the modifications for sequential learning --- in-utterance contrastive loss, position randomization, and convolutional subsampling --- are essential for speech.
2. The authors perform ablations to show the relevance of the projection head vis-a-vis the predictor, and demonstrate they are complementary. This result raises important questions about conclusions drawn about this framework in previous works (like Tian et al. 2021), and their applicability to sequence representation learning.
3. Experimental setup (e.g. hyperparameters) is described in detail. SPIRAL obtains strong WER performance on LibriSpeech with a fraction of the training of wav2vec 2.0. Multi-condition training is shown to be effective at pre-training (versus only at fine-tuning)  to improve noise robustness (~10% relative WER reduction) while maintaining performance on clean speech.
4. The authors have mentioned that they will release the code at the time of publication. This would be useful for the community to extend this direction of research.
5. The paper is clear and easy-to-follow, with sufficient discussion of related work.

Weaknesses:
1. While the proposed model does well on clean LibriSpeech, performance under noisy conditions may be concerning (Section 5.2). In Table 5, when the synthetic noise at test time is matched with training noise (both in noise type and SNR), the WERs are good; but mismatched SNR significantly degrades WER (8.0% to 26.1% for “test-clean”). This suggests that the model may be overfitting to the range of SNRs during training. Second, there are no evaluations on mismatched noise types during train and test, so it is hard to predict the model’s generalizability to other types of noise. Furthermore, the absence of evaluations on real noisy data (such as CHiME-4) raises some questions about the benefits of multi-condition training. These questions are of particular interest since the model is named “perturbation invariant”.
2. There have been several recent works analyzing representation collapse of non-contrastive SSL models (e.g. Tian et al., 2021) which suggest that a predictor on the student branch, and weight decay during training are essential to prevent collapse. The authors mention briefly in Section 2 that the predictor was not sufficient and they had to use the in-utterance contrastive loss with position randomization. Can they suggest reasons why this might be the case?
3. The ablation in Section 5.3.2  indicates that performance degrades significantly when perturbations are applied to the teacher’s input. This again deviates from standard practice in image representation learning where augmentation is applied on the inputs for both the teacher and the student, and (as astutely observed by the authors) is closer in principle to standard self-training. Could the authors suggest why input noise is harmful but computation noise (dropout) works by, for instance, elucidating the link to Gal and Ghahremani (2016)?

Other comments/questions:
1. For multi-condition pre-training, is the noisy input used for both the student and the teacher? If yes, what would happen if the clean input is used for the teacher?
2. For LibriSpeech evaluation under the “low resource” setting, perhaps it would be fairer (although not comparable to Baevski et al., 2020) to exclude the train-clean-100 subset during pre-training, as done in Park et al., 2020.
3. In the first paragraph of Page 2, the authors comment that “SPIRAL also allows to combine with multi-condition training.” This is slightly misleading as there is no inherent limitation in the other models (wav2vec 2.0, HuBERT) mentioned in the previous line that would prevent multi-condition training with those models.
4. Have the authors tried learning from raw waveforms instead of log-mel filterbanks?

Some typos:
* Section 1, para 3: “...learning representation denoising of perturbed data…” →  “...learning denoising representation of perturbed data…”
* Section 3.2, para 1: “predicotr” → “predictor”
* Section 3.5, para 3: “...due limited receptive field.” → “...due to limited receptive field.”



**Summary Of The Paper:**

The paper introduces SPIRAL, a new method for self-supervised pre-training for speech. SPIRAL is based on the teacher-student framework similar to Mean Teacher (Tarvainen and Valpola, 2017) and BYOL (Grill et al., 2020) where the teacher’s weights are updated as a moving average of the student’s weights, but makes additional modifications for sequence tasks like speech:

* An in-utterance contrastive loss is used as the pre-training objective.
* Position randomization of the teacher’s input is used to avoid representation collapse.
* Ablation experiments are done to show that the predictor (which was essential in earlier works like BYOL and SimSiam to avoid collapse) can be replaced with a convolutional projection head without performance degradation.

Empirically, the main contributions of the paper are:
1. Achieving similar/better WER on LibriSpeech compared to wav2vec 2.0 with 35% of the training cost; and
2. Incorporating multi-condition training, which has been used in supervised training in the past, for noise-robust ASR.


**Summary Of The Review:**

While there are some clear limitations in the empirical sense, particularly in the claim that the model learns denoised high-level representations, the paper advances self-supervised learning for speech tasks by adapting the popular teacher-student framework popular in image representation learning. The model achieves WERs similar/better than the popular wav2vec 2.0 architecture while reducing training time and model size. The ablation experiments raise questions about whether the modeling/training strategies deemed essential in that modality are also needed for sequential tasks, although the authors have not addressed these questions directly in this work. Overall, this work (and the promised code release) is likely to lead to new explorations in this direction for speech self-supervised learning.

---

> ### Comment · Reviewer_J87m · 2021-11-22
> **Suggested change to claims in the paper**
>
> While I did recommend a "weak accept", I would have liked to see some discussion from the authors about the weaknesses pointed out in several of the reviews (including mine). In particular, one of the claims made in the abstract is that "we address the problem of noise-robustness that is critical to real-world speech applications." This is an exaggerated claim considering that the model does well only on matched synthetic noise types under the same SNR range as in training. If the authors are not planning to add experiments on mismatched noise and/or on real noisy speech, I would suggest that these claims of "noise robustness" should be tempered down.

---

> > ### Comment · Reviewer_J87m · 2021-11-22
> > **Concern has been addressed**
> >
> > With the addition of results on CHiME-3 real noisy data, I think this is no longer a concern, and the abstract can be retained as is.

---

> ### Author Response · Authors · 2021-11-22
> **Response to Reviewer J87m, CHiME 3 real test set results added.**
>
> Thank you for your valuable assessment of our work and helpful suggestions for improvement. We are sorry for late reply.
> We would like to reply to your comments as follows.
>
> &nbsp;
>
> ---- "Furthermore, the absence of evaluations on real noisy data (such as CHiME-4) raises some questions about the benefits of multi-condition training. "
>
> We have added experiments on real noisy test data from CHiME-3 which is the same data set as CHiME-4. We use the real subset from CHiME 3 test dataset, which is comprised of speech data recorded in real noisy environments (bus, cafe, pedestrian area, and street junction) uttered by actual speakers. The data are recorded with a microphone array composed of multiple microphone channels located at different positions of a tablet. Signal-to-noise ratios (SNR) of the data recorded from different channels are therefore different. The details of microphone configuration can be found in http://spandh.dcs.shef.ac.uk/chime_challenge/chime2015/overview.html.
> We test the models on different microphone channels (mic) of the test set.
> The results are listed below,
>
> | Model                | mic 0 |  mic 5 | mic 1 | mic 2  |
> | :---:                |:----: | :----: | :----:| :---:  |
> | wav2vec 2.0 base          | 23.2  |  56.1  | 68.3  |  98.1  |
> | SPIRAL base               | 24.1  |  52.1  | 58.9  |  92.6  |
> | SPIRAL base FT w/ MCT     | 20.8  |  35.5  | 41.1  |  76.4  |
> | SPIRAL base PT w/ MCT     | 20.8  |  33.6  | 38.5  |  74.0  |
> | SPIRAL base PT & FT w/ MCT| 20.0  |  31.1  | 35.6  |  69.5  |
>
> The results show that multi-condition pre-training significantly improves the performance for real noisy speech for different microphone channels except the mic 0, which is a close-talking microphone with the highest SNR (the least noisy).
> Compared to the model applying MCT solely in fine-tuning, applying MCT both in pre-training and fine-tuning achieves 12.4%/13.3%/9.0% relative WER reduction for mic 1/5/2.
> We note that mic 2 faces backwards to the speaker. SNR of the recordings from mic 2 is the lowest (the noisiest). Therefore, the WER is the highest. We only test for a subset (0,1,2,5) of microphone channels. The other microphones (3,4,6) are on similar relative positions to the speakers on the tablet. We expect the similar performance trends as mic 1 and mic 5.

---

> > ### Comment · Reviewer_J87m · 2021-11-22
> > **Thank you for the CHiME-3 results**
> >
> > Thanks for carrying out the experiments on CHiME-3. It seems that the denoising representations do indeed help even when just used for fine-tuning, compared with the W2V2 baseline. I believe this makes the conclusions about noise robustness much stronger, and I am happy to recommend accept with the addition of these new results.

---

> ### Author Response · Authors · 2021-11-22
> **Response to Reviewer J87m, continue**
>
> We would like to continue our replies to your comments as follows.
>
> &nbsp;
>
> ---- "but mismatched SNR significantly degrades WER (8.0% to 26.1% for “test-clean”). This suggests that the model may be overfitting to the range of SNRs during training."
>
> We presume the more WER degradation is because of the low SNR. We note that the models pre-trained with MCT still achieve significant lower WER compared to other models under lower SNR settings.
>
> &nbsp;
>
> ---- "The authors mention briefly in Section 2 that the predictor was not sufficient and they had to use the in-utterance contrastive loss with position randomization. Can they suggest reasons why this might be the case?"
>
> A possible reason is that learning sequential representation poses unique challenges than learning single global representation. For example, it is important to prevent the positional collapse, while there is no such kind of collapse in global representation learning.
>
> &nbsp;
>
> ---- "..... is closer in principle to standard self-training. Could the authors suggest why input noise is harmful but computation noise (dropout) works by, for instance, elucidating the link to Gal and Ghahremani (2016)?"
>
> In self-training, hard pseudo-labels may contain errors. An alternative is to use soft pseudo-label (https://arxiv.org/pdf/1908.09822.pdf), which contains uncertainty information of the teacher.  Monte Carlo dropout proposed in Gal and Ghahremani (2016) suggests that enabling dropout in a network at inference time induces uncertainty information in output.
>
> &nbsp;
>
> ---- “For multi-condition pre-training, is the noisy input used for both the student and the teacher? If yes, what would happen if the clean input is used for the teacher?”
>
> No, like SpecAugment, additive noise is only applied to the input of the student.
> Teacher always consumes clean input. This is the reason that we call our method as representation denoising.
>
> &nbsp;
>
> ---- "or LibriSpeech evaluation under the “low resource” setting, perhaps it would be fairer (although not comparable to Baevski et al., 2020) to exclude the train-clean-100 subset during pre-training, as done in Park et al., 2020."
>
> Labeled data still can be applied to pre-training stage by ignoring the label. We consider that the setting is still fair. The same data set is applied in which among 960 hours of audio data, 100 hours is labeled.
>
> &nbsp;
>
> ---- "the authors comment that “SPIRAL also allows to combine with multi-condition training.” This is slightly misleading as there is no inherent limitation in the other models (wav2vec 2.0, HuBERT) mentioned in the previous line that would prevent multi-condition training with those models."
> We will update our paper. We will rephase as " We apply multi-condition training with SPIRAL ..."
>
> &nbsp;
>
> ---- "Have the authors tried learning from raw waveforms instead of log-mel filterbanks?"
>
> We leave it for future work. Moreover, log-mel filterbanks allows us to apply SpecAugment as perturbation.
>
> &nbsp;
>
> ---- "Some typos: ..."
>
> We will fix them in the update revision.

---

> > ### Comment · Reviewer_J87m · 2021-11-22
> > **Thanks for the answers**
> >
> > Thank you for clarifying these points.

---

### Official Review · Reviewer_H4es · 2021-11-03

**Correctness:** 3
**Technical Novelty And Significance:** 3
**Empirical Novelty And Significance:** 3
**Recommendation:** 6
**Confidence:** 3

**Main Review:**

The paper describes teacher-student self-supervised training. The teacher generates a representation on clean data. The student matches  the teacher's representation with a perturbed version of the utterance. In-utterance contrastive loss is used to avoid learning a trivial representation. Positional collapse is avoided by random padding before/after the utterance fed to the teacher. The teacher is updated as the moving average of student checkpoints. Results are presented for low-resource and high-resource settings using librispeech and librilight.

The strengths are:
- The paper has an interesting premise, combining ideas from noisy-student training and self-supervised training.
- The results are somewhat comparable to wav2vec 2.0, but at lower training cost.
- The method is robust to unseen noise situations.
- As far as I understand, it does not require separately pre-training the teacher, unlike noisy student training. This could be clarified however.

The weaknesses include:
- The results are better or on par with wav2vec 2.0 in some cases, but not on the whole. The authors mention that the settings are not fully tuned -- it would be interesting to know what the best results are after tuning.
- While there are some ablation studies, some unanswered questions remain. In particular, I am curious how much is gained from additive noise perturbation and from positional randomization.
- In looking at the noisy test results (table 5), there is no additive noise in either pre-training or fine-tuning for wav2vec 2.0. Hence this comparison seems incomplete.

Other questions and notes:
- Related work could include contrastive semi-supervised learning work from Facebook, as another way to combine these two aspects.
- How is the teacher initialized?
- In table 2, I assume training step refers to pre-training steps. How is the optimal number of pre-training steps determined for each method?
- What perturbation settings are included in the wav2vec setup?
- Table 3: noisy student unlabeled data = "LS-860" -- is this a typo?
- Page 7: the-clean -> test-clean (typo)

**Summary Of The Paper:**

The paper describes teacher-student self-supervised training, with a denoising advantage by perturbing the student. It elaborates on techniques to avoid collapse and compares results with wav2vec 2.0.

**Summary Of The Review:**

The paper has an interesting premise and some promising experiments to justify it. On the whole, the method is somewhat comparable to wav2vec 2.0, although it falls short in several cases. This can be made clearer in the abstract and introduction, while highlighting the reduced training cost. I am not convinced that the comparison to wav2vec 2.0 is complete in terms of denoising uses, as there is no discussion of specaug and additive noise in wav2vec 2.0 during pre-training or fine-tuning. Also, some additional ablation studies to understand the benefit of additive noise and positional randomization would strengthen the paper.

---

> ### Author Response · Authors · 2021-11-22
> **Response to Reviewer H4es**
>
> Thank you for your valuable assessment of our work and helpful suggestions for improvement. We would like to reply to your comments as follows.
>
> &nbsp;
>
> ---- "The results are better or on par with wav2vec 2.0 in some cases, but not on the whole. "
>
> We have improved our results after hyper-parameter tuning. The new results are listed in our official comments of our paper. We are still working on improving LM decoding results.
>
> &nbsp;
>
> ---- " I am curious how much is gained from additive noise perturbation and from positional randomization."
>
> First, we want to clarify that when evaluation on the original Librispeech Dataset (section 5.1) , we didn't use additive noise.
>
> In section 5.2, when evaluating noise-robustness of the SPIRAL model, we have given a comparison of models trained with or without additive noise.
> The results show that speech recognition performance of noisy data improves with additive noise perturbation in pre-training. The speech recognition performance for clean data is maintained.
>
> Without positional randomization, the model sometimes achieves very low contrastive loss and high accuracy of the contrastive task of model validation. However, the performance on the downstream speech recognition task is poor. We will probably add a ablations study in the future revisions.
>
> &nbsp;
>
> ---- "In looking at the noisy test results (table 5), there is no additive noise in either pre-training or fine-tuning for wav2vec 2.0. Hence this comparison seems incomplete."
>
> The main focus is to investigate the effects of applying multi-condition pre-training and fine-tuning in SPIRAL. We add Wav2vec 2.0 as baseline without multi-condition pre-training. We do not imply that Wav2vec 2.0 cannot be improved with multi-condition training. We will add a clarification in the updated revision.
>
> &nbsp;
>
> ---- "Related work could include contrastive semi-supervised learning work from Facebook, as another way to combine these two aspects."
>
> We will update our paper in updated revision.
>
> &nbsp;
>
> ---- "How is the teacher initialized?"
>
> The teacher is initialized to the same weights of the student. The weights of the student are randomly initialized.
>
> &nbsp;
>
> ---- "In table 2, I assume training step refers to pre-training steps. How is the optimal number of pre-training steps determined for each method?"
>
> Training step refers to pre-training steps. For SPIRAL, we determine the pre-training steps by the fine-tuning performance of dev-other as a hyper-parameter. The pre-training steps of Wav2vec 2.0 is from the corresponding paper.
>
> &nbsp;
>
> ---- "What perturbation settings are included in the wav2vec setup?"
>
> We use the wav2vec 2.0 model release from the official open source repository.
>
> &nbsp;
>
> ---- "Table 3: noisy student unlabeled data = "LS-860" -- is this a typo?"
>
> It's not a typo. NST use the clean-100 subset as the labeled dataset and the remaining 860-hour data (LS-860) as unlabelled data. We follow the usage of LS-860 in wav2vec 2.0 paper. We will add a clarification in the future revisions.
>
> &nbsp;
>
> ---- "Page 7: the-clean -> test-clean (typo)"
>
> We will fix it in the updated revision.

---

### Official Review · Reviewer_GA9R · 2021-11-03

**Correctness:** 4
**Technical Novelty And Significance:** 3
**Empirical Novelty And Significance:** 3
**Recommendation:** 8
**Confidence:** 3

**Main Review:**

Strengths:

The paper is overall well written, and presents a good set of experimental results. The baselines are strong (for what I can tell), the approach is well motivated and explained well. The authors promise to release code upon acceptance of the paper, which should allow readers to verify results and build on top of it. Pre-trained self-supervised audio feature extractors have the potential to improve speech recognition in many domains (e.g. low resource multi-lingual speech recognition) and reducing compute is a critical step in that direction.

Weaknesses:

Some of the experimental results are a bit early? I would like to see results with LM rescoring after the hyper parameters have been optimized, so that results are more comparable (and hopefully consistent)? Similarly with mixed precision training. Would it be possible to get rid of the convolutional layers and build a model that is based entirely on self-attention? This should be even more efficient on GPU?

Update: the additional tests on Chime data, and the updated decoding results address these concerns. I am updating my assessment.

**Summary Of The Paper:**

The paper presents a novel approach to self-supervised speech representation learning, which promises to be simpler than existing methods such as wav2vec 2.0. The approach is inspired by the BYOL approach from CV, and is shown to be indeed largely as effective as wav2vec 2.0, while being significantly more efficient during training.

**Summary Of The Review:**

Making self-supervised speech representations easier to train is a significant contribution, and this paper presents a viable approach to doing so. Some results feel a bit preliminary, but the paper is well written and the authors may have updated results but the time of the conference, and they will release code (plus presumably models?). My recommendation is thus for accept.

---

> ### Author Response · Authors · 2021-11-22
> **Response to Reviewer GA9R**
>
> Thank you for your valuable assessment of our work and helpful suggestions for improvement.
> We would like to reply to your comments as follows.
>
> ---- "Some of the experimental results are a bit early? I would like to see results with LM rescoring after the hyper parameters have been optimized, so that results are more comparable (and hopefully consistent)? "
>
> We have improved our results after hyper-parameter tuning. The new results are listed in our official comments of our paper. We are still working on improving LM decoding results.
>
> &nbsp;
>
> ---- "Similarly with mixed precision training."
>
> Due to limited time, we will probably leave it as future work for model training procedure optimization.
>
> &nbsp;
>
> ---- "Would it be possible to get rid of the convolutional layers and build a model that is based entirely on self-attention? This should be even more efficient on GPU?"
>
> This is possible as there are many works on applying self-attention only Transformer models for speech recognition. Adding convolutional layers further improves ASR results, such as Conformer (https://arxiv.org/abs/2005.08100).

---

### Official Review · Reviewer_g6zH · 2021-11-03

**Correctness:** 4
**Technical Novelty And Significance:** 3
**Empirical Novelty And Significance:** 3
**Recommendation:** 8
**Confidence:** 4

**Main Review:**

The paper is well written with the following strengths.
(1) The proposed method is self-supervised learning with the teacher-student framework that is an extension to Mean Teacher (MT) and Boostrap Your Own Latent (BYOL).
(2) The proposed method is a novel one that aims to learn the representation denoising of perturbed data with the teacher-student framework.  In addition, the proposed method can also be combined with multi-condition training to improve the noise-robustness.
(3) The motivation of the method sounds quite reasonable which aims to learn the representation that is invariant to perturbation so that the learnt representation is a high-level one (e.g. carrying content information) to enhance the downstream speech applications.
(4) An in-utterance contrastive loss (proposed by Chopra et al. (2005) is adopted to avoid the model collapse problem.
(5) Position randomization technique is further introduced to prevent the positional collapse problem.
(6) A gradual down-sampling strategy has been adopted to train the SPIRAL model to reduce the computation cost.
(7) Plenty of experiments have been conducted to evaluate the performance of the proposed method.
(8) The proposed method can achieve competitive or better results than wav2vec 2.0 whereas with significantly less training cost.



**Summary Of The Paper:**

The paper proposes a new speech pre-training approach named SPIRAL.  The proposed method is trained by learning representation denoising of perturbed data with the teacher-student framework in a self-supervised manner.  The motivation of the method is to learn the representation that is invariant to perturbation so that the learnt representation is a high-level one (e.g. carrying content information) that can enhance the downstream speech applications.  Compared to the state-of-the-art speech pre-training method wav2vec 2.0, the proposed method can achieve competitive or better results but with a significant reduction of the training cost.  The proposed method is also able to deal with the noise-robustness problem.

**Summary Of The Review:**

Based on the above main review (especially the strengths of the paper), although some of the ideas have been borrowed from previous work (e.g. MT, BYOL, contrastive loss, etc), the paper has proposed extensions to these work by considering the sequential applications in speech processing.  And the relations with the previous work have been clearly discussed in section 2.  Hence, I think the paper could be accepted for publication on ICLR.

---

> ### Author Response · Authors · 2021-11-22
> **Response to Reviewer g6zH**
>
> Thank you for your positive review of our work. We have improved our results after hyper-parameter tuning. The new results are listed in our official comments of our paper. We have replied the comments of other reviewers and will update our paper accordingly.

---

### Official Review · Reviewer_fPpa · 2021-11-03

**Correctness:** 4
**Technical Novelty And Significance:** 3
**Empirical Novelty And Significance:** 3
**Recommendation:** 8
**Confidence:** 5

**Main Review:**

The goal of perturbation or augmentation invariance is well motivated in speech recognition, and also ML/ representation learning more broadly.  The Librispeech 960 numbers are quite strong.

* The results using synthetic noisy data are less convincing than naturally occurring noise.  This is especially important since the synthetic noise is used during training as well.
* In figure 1 it appears that the positional padding is applied equally to both the input and output of the teacher.  This is not as clear from the description in Section 3.1 If this is the case it would be helpful to expand on the description.
* Section 3.3: SpecAugment typically masks both time and frequency bands.  It is not clear if this is done here, or if the masking is performed on either time or frequency.  Assuming both time and frequency are masked, is the area masked by both time and frequency replaced with zeros or gaussian noise?
* Section 3.5: What is the rationale for using CTC ASR decoding rather than say an attention decoder, or RNN-T model?
* Section 4.2 it would be useful to have more insight into why SPIRAL trains faster than wav2vec 2.0.  Is this due to implementation, or is there something fundamental to these algorithms that makes SPIRAL faster?
* Section 4.2: It might be useful to compare these results to SpeechStew
* Section 5.3.1: it is interesting that other approaches are more robust to perturbing both teacher and student inputs, while SPIRAL requires clean teacher inputs. More thorough discussion of this would be helpful, though space is tight.

Shorter notes:
* Section 2: the description of Liang et al (2018) is notably brief compared to others.  The comparison of perturbation invariance in ASR training seems like a worthwhile comparison to invariance during pretraining.
* Table 3: only row 2 includes 3 significant figures, the rest includes 2. it would be better to be consistent here.  (Same comment in tables 4. and 5)
* Table 5: it would be better to organize the columns in by increasing or decreasing SNR.
* Typo in Section 5.3.2 where "When the predictor is removed, We observe" the "We" shouldn't be capitalized.

**Summary Of The Paper:**

This paper describes a self supervision training approach to pretrain speech encoders.  This is related to recent work on wav2vec 2.0, and SimCLR.  The contrastive loss is similar to wav2vec 2.0, while the invariance to perturbation (i.e. augmentation) is akin to SimCLR.
Performance is competitive with wav2vec 2 on Librispeech.

**Summary Of The Review:**

This is a compelling paper. it's a well motivated and technically sound perspective on self-supervised training.  The performance on Librispeech 960 is quite strong.  The performance in noise conditions is strong, but would be more convincing if shown on more actual rather than synthetic noise.

---

> ### Author Response · Authors · 2021-11-22
> **Response to Reviewer fPpa**
>
> We thank the reviewer for the detailed review and encouraging feedback. Here are our responses to the comments.
>
> &nbsp;
>
> ---- “The results using synthetic noisy data are less convincing than naturally occurring noise. This is especially important since the synthetic noise is used during training as well."
>
> First, we would like to clarify that the noise data used in training and test are from different subsets of the noise dataset.
>
> Second, we have added experiments on the real noisy test data from CHiME-3. The results suggest that our method is still effective for real noisy data. The details and the results are posted in the response to Reviewer J87m, and will be included in the updated version of our paper.
>
> &nbsp;
>
> ---- "In figure 1 it appears that the positional padding is applied equally to both the input and output of the teacher. "
>
> The output corresponding the positional padding is excluded from contrastive loss computation. We will update the figure in the paper.
>
> &nbsp;
>
> ---- "SpecAugment typically masks both time and frequency bands. It is not clear if this is done here, ..."
>
> In spiral pre-training,  both time and frequency bands are masked. Time mask is applied after frequency mask. For regions with both time and frequency mask, we fill in the values of the regions with Gaussian noise.
>
> &nbsp;
>
> ---- "What is the rationale for using CTC ASR decoding rather than say an attention decoder, or RNN-T model?"
>
> We use CTC objective for comparability to Wav2vec 2.0. We expect further performance improvements with auto-regressive objectives. We leave it as future work.
>
> &nbsp;
>
> ---- "it would be useful to have more insight into why SPIRAL trains faster than wav2vec 2.0."
>
> One possibility is that masked prediction methods including Wav2vec 2.0 only calculate loss for the masked regions, while SPIRAL is based on a teacher-student framework, the loss is calculated for the whole sequence, and using latent representation of the teacher as the targets can provide more rich information.
> Due to limited time and scope, we will probably leave it for future work.
>
> &nbsp;
>
> ---- "It might be useful to compare these results to SpeechStew"
>
> SpeechStew is great work. We are also interesting to investigate how SPIRAL scales with more diverse data and bigger model size. Due to limited time and scope, we will probably leave it for future work.
>
> &nbsp;
>
> ---- "other approaches are more robust to perturbing both teacher and student inputs, while SPIRAL requires clean teacher inputs. More thorough discussion of this would be helpful, though space is tight."
>
> In the related work, we view SPIRAL as an extension of self-training methods to pre-training. Feeding clean inputs to teacher seems a natural choice.
>
> &nbsp;
>
> ---- "The comparison of perturbation invariance in ASR training seems like a worthwhile comparison to invariance during pre-training."
>
> This is a worthwhile investigation. We will probably leave it for future work.
>
> &nbsp;
>
> ---- "Table 3: only row 2 includes 3 significant figures, the rest includes 2. "
>
> For the results from previous works, we directly refer the published results, which leads to different significant figures.
>
> &nbsp;
>
> ---- "Table 5: it would be better to organize the columns in by increasing or decreasing SNR."
>
> We will update in the revised version.
>
> &nbsp;
>
> ---- "Typo in Section 5.3.2 where "When the predictor is removed, We observe" the "We" shouldn't be capitalized."
>
> The typo will be fixed in the revised version.

---

### Author Response · Authors · 2021-11-22
**Result improvements after tuning a small number of hyper-parameters and fixing a bug**

We thank the reviewers for your valuable assessment of our work and helpful suggestions for improvement.
In this response, we give an overview of  improvements of our work with hyper parameter tuning and bug fix after the initial submission.

## Result improvements after tuning a small number of hyper-parameters and fixing a bug

We list the improvements in following tables. In the tables, the numbers in parentheses are the absolute WER reduction compared to the old models.

### Low resource setup improvement

| Model               | LM      | dev clean  | dev other   | test clean | test other  |
| :---:               |:----:   |:----:      |:----:       |:----:      |:----:       |
| wav2vec 2.0 base    | -       | 6.1        | 13.5        | 6.1        | 13.3        |
| SPIRAL base   | -       | 6.2        | 11.7        | 6.0        | 11.9        |
| SPIRAL base (updated)   | -       | 5.5 (-0.7) | 11.1 (-0.6) | 5.4 (-0.6) | 11.2 (-0.7) |
| wav2vec 2.0 large   | -       | 3.3        | 6.5         | 3.1        | 6.3         |
| SPIRAL large  | -       | 3.7        | 6.5         | 3.6        | 6.7         |
| SPIRAL large (updated)  | -       | 3.3 (-0.4) | 5.9 (-0.6)  | 3.3 (-0.3) |  6.3 (-0.4) |

### High resource setup improvement:

| Model               | LM      | dev clean   | dev other  | test clean | test other  |
| :---:               |:----:   |:----:       |:----:      |:----:      |:----:       |
| wav2vec 2.0 large   | -       | 2.1         | 4.5        | 2.2        | 4.5         |
| SPIRAL large   | -       | 2.2         | 4.8        | 2.4        | 4.9         |
| SPIRAL large (updated)  | -       | 2.1 (-0.1)  | 4.3 (-0.5) | 2.2 (-0.2) | 4.6 (-0.3)  |

Next, we describe how we achieve these improvements by tuning a small number of hyper parameters and bug fix.

### Tuning the rate of teacher's weight updates for SPIRAL large pre-training.

We get better performance by using a larger rate of teacher's weight updates of SPIRAL, i.e. , using a higher weight update rate.
Specifically, in previous version, the value of $\alpha_t$ is increased from $0.995$ to $1.0$.
After doing some parameter sweeping. We choose to increase the value of $\alpha_t$ from $0.990$ to $0.999$.
We decide not to further tuning other hyper parameters of pre-training for time-constraint, although there is possibly further tuning space.

### SpecAugment tuning of SPIRAL base and large fine-tuning.

We found that using weaker SpecAugment with fewer number of masks on time and frequency domains improves all of our models.
We tune the SpecAugment configuration on dev-other.
Since we use the same SpecAugment configuration in our ablation study, we re-run all the ablation experiments. We observe consistent improvement with the new SpecAugment configuration.


## Better frozen model performance after fixing a bug of frozen tuning

We also fix a bug for fine-tuning of the frozen models.
The bug is that we turned off the gradient computation of the pre-trained weights, but weight decay is still applied to the pre-trained weights.
After this bug is fixed, our best frozen model achieves lower WER than Jasper (https://arxiv.org/abs/1904.03288) on Lirispeech. This demonstrates that SPIRAL learns useful high-level features in pre-training and can serve as a good feature extractor.

| Model                 | test clean | test other |
| :----:                |:----:      |:----:      |
|Jasper                 |       3.86 |      11.95 |
|Spiral frozen          |       4.4  |        7.4 |
|Spiral frozen (updated)|       3.5  |        6.4 |

---

### Author Response · Authors · 2021-11-23
**Current results of LM decoding**

We spend lots of efforts to improve SPIRAL with LM decoding, as the performance is under our expectation in some cases. We have updated our decoding library to fix a reported CTC decoding bug (https://github.com/flashlight/flashlight/commit/9ef0d588b3fbcae11b65c12b689787964b1e8b90).

During the process, we found a mistake in our previous test that the lexicon for LM decoding is built from words of the test set (This problem does not affect decoding without LM which is lexicon-free. The models directly output and combine subword units). This is an unfair advantage of SPIRAL to the baseline when decoding with LM. When we correct this mistake by re-building lexicon from vocabulary of the LM, we observe significant performance degradation. The current WERs with LM are worse than those in the initial submission, although the performance of the new model is improved without LM. When LM is not used, our models performs on par or better than wav2vec 2.0. However, the current performance of SPIRAL with LM decoding is worse than wav2vec 2.0.

We observe a few mismatches with our implementation and the Transformer LM. The transformer LM are word-based, but the SPIRAL is operating in subword-based sentence-piece. As the mapping of words to subwords is not unique, this could hinder the beam search performance. Due to limited time, we only use the same word-based LM as wav2vec 2.0. We search the LM decoding hyper-parameter at step-size of 0.1, while the step-size of hyper-parameter search of wav2vec 2.0 is 0.01.

We still consider SPIRAL a significant work, as the models perform well without LM, and the excellent performance of SPIRAL with frozen model parameters.

We are continuing to improve the LM decoding.  We will build a subword-level LM for further performance improvement. We will keep updating the results in future revisions of our paper.

We would like to report the current results of LM decoding (corrected lexicon) in the following tables.

## Low resource setup with LM

| Model               | LM      | dev clean  | dev other   | test clean | test other  |
| :---:               |:----:   |:----:      |:----:       |:----:      |:----:       |
| wav2vec 2.0 base    | Transf. | 2.2        | 6.3         | 2.6        | 6.3         |
| SPIRAL base  | Transf. | 2.9        | 6.8         | 3.3        | 6.9         |
| SPIRAL base (updated)   | Transf. | 2.9          | 6.8           | 3.2          |  7.2            |
| wav2vec 2.0 large   | Transf. | 1.9        | 4.0         | 2.0        | 4.0         |
| SPIRAL large | Transf. | 1.9        | 4.2         | 2.3        | 4.1         |
| SPIRAL large (updated)  | Transf. | 2.4          |  4.5           |  2.5          | 4.8           |

## High resource setup with LM

| Model               | LM      | dev clean  | dev other   | test clean | test other  |
| :---:               |:----:   |:----:      |:----:       |:----:      |:----:       |
| wav2vec 2.0 large   | Transf. |  1.6        |  3.0         | 1.8        | 3.3         |
| SPIRAL large (original) | Transf. | 1.4        |  3.2         | 1.7        | 3.2         |
| SPIRAL large (updated)  | Transf. | 1.7          | 3.5           | 1.9          | 3.7           |

---

### Author Response · Authors · 2021-11-23
**Updated manuscript with new CHiME-3 results, updated experimental results, and clarifications**

We thank the reviewers for your valuable assessment of our work and helpful suggestions for improvement.

In the updated manuscript, we add new experiments on CHiME-3 for noise-robust evaluation. We also update the results after hyper-parameter tuning. The results of LM decoding is also updated. Based on suggestions from reviewers, we have added some clarifications.
The details of the update can be found in our previous comments.

---

### Decision · Program_Chairs · 2022-01-20

**Decision:**

Accept (Poster)

**Comment:**

This paper proposed a self-supervised speech pre-training approach, by the name of SPIRAL, to learning perturbation-invariant representations in a teacher-student setting.  The authors introduced a variety of techniques to improve the performance and stabilize the training.  Compared to the popular unsupervised learning model wav2vec 2.0, better WERs were reported using SPIRAL with a reduced training cost.  All reviewers considered the work solid with sufficient novelty but also raised concerns regarding the generalization under unseen real-world noisy conditions and missing decoding details.  The authors responded with new Chime-3 results  and updated LM decoding results.  The new results show that, after a bug fix, SPIRAL can outperform wav2vec 2.0 when no external LM is used.

Overall the proposed approach is technically novel.  The experiments are extensive and the results are compelling. In addition, the training time can be significantly reduced compared to wav2vec 2.0. All reviewers are supportive.  So I would recommend accept.